# CmNAC73 Mediates the Formation of Green Color in Chrysanthemum Flowers by Directly Activating the Expression of Chlorophyll Biosynthesis Genes *HEMA1* and *CRD1*

**DOI:** 10.3390/genes12050704

**Published:** 2021-05-08

**Authors:** Jing Luo, Huan Wang, Sijia Chen, Shengjing Ren, Hansen Fu, Ruirui Li, Caiyun Wang

**Affiliations:** 1Key Laboratory for Biology of Horticultural Plants, Ministry of Education, College of Horticulture and Forestry Sciences, Huazhong Agricultural University, Wuhan 430070, China; ljcau@mail.hzau.edu.cn (J.L.); wanghuan149808@163.com (H.W.); csj13476121778@163.com (S.C.); renshengjing8479@163.com (S.R.); fhs4242021@163.com (H.F.); lrr510819@163.com (R.L.); 2Key Laboratory of Urban Agriculture in Central China, Ministry of Agriculture, Wuhan 430070, China

**Keywords:** chrysanthemum, transcriptome, green color, chlorophyll, NAC

## Abstract

Chrysanthemum is one of the most beautiful and popular flowers in the world, and the flower color is an important ornamental trait of chrysanthemum. Compared with other flower colors, green flowers are relatively rare. The formation of green flower color is attributed to the accumulation of chlorophyll; however, the regulatory mechanism of chlorophyll metabolism in chrysanthemum with green flowers remains largely unknown. In this study, we performed Illumina RNA sequencing on three chrysanthemum materials, *Chrysanthemum vestitum* and *Chrysanthemum morifolium* cultivars ‘Chunxiao’ and ‘Green anna’, which produce white, light green and dark green flowers, respectively. Based on the results of comparative transcriptome analysis, a gene encoding a novel NAC family transcription factor, CmNAC73, was found to be highly correlated to chlorophyll accumulation in the outer whorl of ray florets in chrysanthemum. The results of transient overexpression in chrysanthemum leaves showed that CmNAC73 acts as a positive regulator of chlorophyll biosynthesis. Furthermore, transactivation and yeast one-hybrid assays indicated that CmNAC73 directly binds to the promoters of chlorophyll synthesis-related genes *HEMA1* and *CRD1*. Thus, this study uncovers the transcriptional regulation of chlorophyll synthesis-related genes *HEMA1* and *CRD1* by CmNAC73 and provides new insights into the development of green flower color in chrysanthemum and chlorophyll metabolism in plants.

## 1. Introduction

Color is an important ornamental trait in plants. Flower color plays an important role in the co-evolution of plants and insects, as flowers use their color to attract insects or birds for pollination [1]. In the tropics, plants often use bright colors to protect themselves from herbivores. In addition, at high altitudes, flowers usually accumulate high levels of anthocyanin, which protects the plants from damage by ultraviolet (UV) radiation [2]. In addition to anthocyanins, flower color is also influenced by carotenoids and chlorophyll. In nature, green flowers are relatively rare, which was disadvantageous for the recognition of insect pollinators. Since chlorophyll plays an important role in photosynthesis, a number of studies have focused on the mechanism of chlorophyll metabolism and regulation in leaves, which is of great importance in delaying leaf senescence and increasing the crop yield [3].

Chlorophyll biosynthesis and degradation are complex processes involving many enzymes. In the chlorophyll biosynthesis cycle, the conversion of glutamyl-tRNA (Glu-tRNA) to chlorophyll-a involves 14 steps, catalyzed by Glu-tRNA reductase (HEMA), glutamate 1-semialdehyde aminotransferase (GSA), porphobilinogen synthase (HEMB), hydroxymethylbilane synthase (HEMC), uroporphyrinogen III synthase (HEMD), uroporphyrinogen decarboxylase (HEME), coproporphyrinogen oxidative decarboxylase (HEMF), protoporphyrinogen oxidase (HEMG), Mg-chelatase D subunit /H subunit/I subunit (CHLD/H/I), Mg-protoporphyrin IX methyltransferase (CHLM), Mg-protoporphyrinogen IX monomethylester cyclase (CRD), protochlorophyllide reductase (PORA/B/C), divinyl reductase (DVR) and chlorophyll synthase (CHLG). In the chlorophyll cycling process, chlorophyllide a oxygenase (CAO), chlorophyll(ide) b reductase (NYC1) and 7-hydroxymethyl chlorophyll-a reductase (HCAR) catalyze the conversion between chlorophyll-a and chlorophyll-b. Additionally, chlorophyll degradation is catalyzed by STAY-GREEN (SGR), pheophytin pheophorbide hydrolase (PPH), pheophorbide a oxygenase (PAO) and red chlorophyll catabolite reductase (RCCR) [4]. Mutations in genes involved in chlorophyll synthesis, such as *HEMA*, significantly inhibit chlorophyll synthesis and cause leaf yellowing [5], whereas mutations in the chlorophyll degradation gene *SGR* results in the stay-green phenotype where leaves remain green for a long time [6].

Chlorophyll metabolism is influenced by many factors, such as developmental cues, light, hormone level and nutrition. Under moderate stress, the rate of chlorophyll breakdown is greater than that of chlorophyll synthesis in old leaves, and the nutrients are transported to other organs to maintain normal growth [7]. Light plays a complex role in chlorophyll metabolism. Dark conditions induce chlorophyll degradation, which is mediated by the PHYTOCHROME-INTERACTING FACTORS (PIFs) [8]. Low light intensity promotes chlorophyll accumulation to compensate for the lack of light [9]. By contrast, intense light stimulates mitochondrial respiration, leading to the overaccumulation of NADPH, which inhibits chlorophyll synthesis [10].

Hormone signals also play a key role in the regulation of chlorophyll metabolism. Abscisic acid (ABA) and jasmonic acid (JA) promote chlorophyll degradation [11], whereas Cytokinin (CTK), Gibberellin (GA) and auxin promote chlorophyll synthesis [12]. The role of ethylene in chlorophyll metabolism is complex. During the transition from skotomorphogenesis to photomorphogenesis, ETHYLENE INSENSITIVE 3 (EIN3) promotes chlorophyll synthesis by directly binding to the promoters of *PORA* and *PORB* [13], whereas, in the leaves of adult plants, ethylene promotes chlorophyll degradation by activating *NYC1* and *PAO* genes [14].

Transcription factors play important roles in regulating the expression of chlorophyll synthesis and degradation-related genes in response to light and hormone signals. In *Arabidopsis thaliana*, several NAC (short for No Apical Meristem [NAM], Arabidopsis Transcription Factor [ATAF] and Cup-shaped Cotyledon [CUC]) transcription factors play important roles in ABA-regulated leaf senescence. Most of the NAC family transcription factors, including ORE1/NAC2, ANAC016, ANAC019, NAC29, ANAC046, ANAC055, ANAC072 and NAC092, promote chlorophyll degradation and leaf senescence [15,16,17]. By contrast, in wheat (*Triticum aestivum* L.), the *NAC* gene *TaNAC-S* negatively regulates leaf senescence, and overexpression of *TaNAC-S* leads to the stay-green phenotype and higher yield [18]. However, whether *TaNAC-S* is involved in chlorophyll synthesis remains unknown. In chrysanthemum (*Chrysanthemum morifolium* Ramat.), the expression of *CONSTANS-like 16* (*COL16*) is highly associated with the chlorophyll content in different cultivars. In petunia (*Petunia ×*
*hybrida*), overexpression of the *COL16* homolog, *PhCOL16a*, significantly enhanced the synthesis of chlorophyll [4,19], although the underlying mechanism remains unknown. 

Chrysanthemum has been cultivated in China for more than 1500 years [4] and is considered an important flower around the world. Green chrysanthemums are very rare but also very popular in the market. However, the molecular mechanism underlying the development of green flower color in chrysanthemum remains unclear. In this study, three chrysanthemum materials with different flowers colors, including *Chrysanthemum vestitum* (white flowers) and *C. morifolium* cultivars ‘Chunxiao’ (light green flowers) and ‘Green anna’ (dark green flowers), were analyzed with Illumina RNA sequencing (RNA-seq) performed by Novogene Co. Ltd. (Beijing, China), and the key transcription factor which mediates the synthesis of chlorophyll were be characterized, through transient overexpression of candidate genes in chrysanthemum leaves, transactivation of chlorophyll synthesis-related genes, and yeast one-hybrid assay. This work will help to uncover the transcriptional regulation of chlorophyll synthesis-related genes and provides a better understanding of the formation of green color in chrysanthemum flowers and the metabolism of chlorophyll in plants.

## 2. Materials and Methods

### 2.1. Plant Materials

Wild chrysanthemum (*Chrysanthemum vestitum*) and six cultivars of hybrid chrysanthemum (*Chrysanthemum morifolium*) with green flowers, ‘Green peony’, ‘Green gemstone’, ‘Chunxiao’, ‘Greenlizard’, ‘Green anna’ and ‘Lv Dingdang’, were grown in the greenhouse of Huazhong Agricultural University (Figure 1). The outer whorl of ray florets was sampled from the fully opened flowers of *C. vestitum* and *C. morifolium* cultivars ‘Chunxiao’ and ‘Green anna’ at the same time, with three biological replicates. The samples were quickly wrapped in foil, frozen in liquid nitrogen, transported to the laboratory and stored in a −80 °C freezer until needed for RNA extraction.

To conduct transient gene overexpression assay and ABA treatment, cut flowers of ‘Lv Dingdang’ were purchased from Nanhu flower market and transported to the laboratory within one hour. Flower stems were cut to a length of 40 cm and then placed in vases containing distilled water. 

### 2.2. Determination of Chlorophyll, Carotenoid and Flavonoid Contents 

The total chlorophyll content of ray florets was determined as described by Fu et al. [4]. Total carotenoid and total flavonoid contents were determined according to the methods of Potosí-Calvache et al. [20] and Cao et al. [21], respectively.

### 2.3. RNA Extraction and RNA-seq

The total RNA was isolated from the ray florets of all three accessions (three biological replicates per accession) using the EASYspin Plant RNA Kit (Aidlab Biotech, Beijing, China). The nucleic acid was quantified with nanodrop 2000 (Thermo Fisher Scientific, Waltham, MA, USA). The integrity of RNA was evaluated with agarose gel electrophoresis and Agilent 2100 (Agilent Technologies, Santa Clara, CA, USA). RNA-seq was performed by Novogene Co. Ltd. (Beijing, China) using the Illumina HiSeq 2500 platform. After removing adaptor and low-quality sequences (Q value < 20), clean data were assembled with the Trinity software (version: r20140413p1), and the assembled transcripts were used as the reference sequence. The clean data of each sample were mapped onto the reference sequence using the RNA-Seq tools of the Expectation-Maximization (RSEM) software (v1.2.15), and read counts were generated. Gene annotation was performed based on seven databases, NCBI non-redundant protein sequence (NR), NCBI nucleotide sequence (NT), Protein Family (Pfam), Eukaryotic Orthologous Groups (KOG), SWISS-PROT, Kyoto Encyclopedia of Genes and Genomes (KEGG) and Gene Ontology (GO), using diamond (v0.8.22), blast (v2.2.28+), KAAS (r140224), hmmscan (HMMER3) and blast2go (b2g4pipe_v2.5) tools. Differentially expressed genes (DEGs) were analyzed using DESeq with the following parameters: fold-change (FC) > 2 or FC < 0.5, and adjusted *p*-value (*padj*) < 0.05. GO and KEGG analyses of DEGs were carried out using the GOSeq (1.10.0) and KOBAS (v2.0.12) software (*padj* < 0.05).

### 2.4. Quantitative Real-time PCR (qRT-PCR)

In total, 1 μg RNA of each sample was used to synthesize cDNA with the cDNA Synthesis SuperMix (AE311-03; TransGen Biotech, Beijing, China), according to the manufacturer’s instructions. Subsequently, qRT-PCR was performed on LightCycler 96 (Roche, Basel, Switzerland) using the cDNA template and sequence-specific primers (Appendix A). All reactions were performed with three biological replicates. The thermocycling program was as follows: 95 ℃ for 5 min, followed by 40 cycles of three steps (95 ℃ for 15 s, 60 ℃ for 30 s, and 72 ℃ for 30 s). In the end, there was a melting curve analysis: 95 ℃ for 15 s, 60 ℃ for 60 s, and 95 ℃ for 15 s, which was used to ensure the specificity of the amplified product. The 2^−ΔΔCt^ method [22] was used to calculate relative gene expression levels. The chrysanthemum *Ubiquitin* (*CmUBI*) gene (accession no. EU862325) was used as the internal reference.

### 2.5. Gene Cloning and Vector Construction

The *CmNAC73* gene sequence was obtained from the chrysanthemum genome [23], and the transcriptome data generated in this study and corrected with high-fidelity Phusion DNA Polymerase (Thermo Fisher Scientific, Waltham, MA, USA). The promoter sequences of chlorophyll synthesis-related genes were cloned according to the chrysanthemum genome, combined with the FPNI-PCR method, as described by Wang et al. [24]. 

To generate the *CmNAC73* overexpression (OE) vector, the open reading frame (ORF) of *CmNAC73*, harboring the restriction sites Hind III and Kpn I, was amplified by PCR. The PCR product was digested and cloned into the pSuper1300 vector, which was derived from pCAMBIA1300 by the research group of Dr. Zhizhong Gong (China Agricultural University). 

To analyze promoter activity, the promoters of candidate genes carrying the Hind III and BamH I restriction sites were amplified by PCR. The PCR products were digested and ligated, and the promoter with the appropriate restriction site was cloned into the PBI121 vector.

### 2.6. Transient Expression of CmNAC73 in Chrysanthemum Leaves

*Agrobacterium tumefaciens* GV3101 cells carrying the pSuper1300 empty vector or pSuper::CmNAC73 were harvested by centrifugation and resuspended in infiltration buffer (10 mM MgCl_2_, 150 µM acetosyringone [As], and 10 mM 2-morpholinoethanesulfonic acid [MES], pH 5.6), and the concentration of the cell suspension was adjusted to obtain an optical density (OD_600_) of 0.8. Subsequently, the *A. tumefaciens* suspension was incubated in the dark, without shaking, for 2 h. Leaves of all three chrysanthemum accessions were agroinfiltrated under vacuum (0.5 atm). After agroinfiltration, the leaves were washed with deionized water and placed in a Petri dish lined with wet filter paper. The Petri dish was incubated in the dark at 8 °C for 3 days and then at 23 °C, 60% relative humidity and 5000 lux light intensity (time set to 0 days). The leaves were photographed every 2 days and used for RNA extraction after 4 days.

### 2.7. Transient Transactivation Assay

The pSuper::CmNAC73 effector plasmid, PBI121 reporter plasmid harboring the promoter of chlorophyll synthesis related genes, and corresponding empty vectors were separately transformed into *A. tumefaciens* strain GV3101. After overnight culture in Luria-Bertani (LB) broth, cells were collected by centrifugation at 5000× *g* for 8 min, and resuspended in infiltration solution (10 mM MES, 10 mM MgCl_2_, and 20 μM As [pH5.6]) (OD_600_ = 1.6). Cells transformed with the effector and reporter plasmids were mixed at a 1:1 ratio, incubated at room temperature without shaking for 2 h, and then infiltrated into the abaxial surface of tobacco (*Nicotiana benthamiana*) leaves using a 1-mL needle-less syringe. The plants were incubated in the dark at 23 °C and 40–60% relative humidity for 3 days. Subsequently, the agroinfiltrated leaves were collected, immediately frozen in liquid nitrogen, and stored at −80 °C.

### 2.8. Determination of β-glucuronidase (GUS) Activity

The agroinfiltrated *N. benthamiana* leaves were ground in liquid nitrogen, and soluble proteins were extracted with the extraction buffer containing 0.05 M phosphate-buffered saline (PBS; pH 7.0), 0.1% sodium N-lauroylsarcosine, 10 mM EDTA (pH 8.0), 1/5 volume of methanol, 1/1000 volume of Triton X-100 and 1/1000 volume of β-mercaptoethanol. The protein extracts were centrifuged at 8000× *g* for 10 min at 4 °C. The protein concentration was determined with the Bradford method [25]. To measure GUS activity, 20 μL of the purified protein was mixed with 180 μL of 2 mM 4-methylumbelliferyl β-d-glucuronide hydrate (4-MUG; substrate) dissolved in extraction buffer. Then, half the volume of the mixture (100 μL) was combined with 900 μL of 0.2 M Na_2_CO_3_ to immediately terminate the reaction, while the remaining 100 μL was incubated at 37 °C for 15 min before the reaction was terminated. The fluorescence of 4-methylumbelliferyl (4-MU; reaction product) was measured using a fluorescence spectrophotometer (F-4500; Hitachi, Tokyo, Japan) at 365-nm excitation and 455-nm emission. GUS activity was defined as the amount of 4-MU produced per mg protein per min (μM 4-MU mg^−1^ protein min^−1^).

### 2.9. Yeast One-Hybrid Assay

The yeast one-hybrid assay was performed as described previously [26]. Briefly, the *CmNAC73* ORF was cloned into pGADT7, and the promoters of chlorophyll synthesis-related genes were cloned into pHIS2.1. The recombinant pGADT7 and pHIS2.1 vectors were co-transfected into yeast (*Saccharomyces cerevisiae*) strain Y187. The transformed yeast cells were cultured on a synthetic-defined medium lacking leucine and tryptophan (SD/-Leu-Trp). Positive clones were verified by PCR and transferred to SD medium lacking Leu, Trp and histidine (SD/-Leu-Trp-His), supplemented with 0–80 mM 3-aminotriazole (3-AT).

### 2.10. Statistical Analysis

Statistical analysis of the data was carried out with SPSS 22.0 (IBM, Armonk, NY, USA). A two-tailed Student’s *t*-test (* *p* < 0.05, ** *p* < 0.01) was used for the comparison of two groups, and one-way analysis of variance (ANOVA) with Duncan’s multiple range tests (*p* < 0.05) was used for comparison of multiple groups.

## 3. Results

### 3.1. Pigment Contents in Chrysanthemum Materials

To understand the basis of green flower color in chrysanthemum, we determined the contents of major pigments in *C. vestitum* and six *C. morifolium* cultivars. From *C. vestitum* to chrysanthemum ‘Lv Dingdang’, the chlorophyll and carotenoid contents of flowers increased gradually, consistent with the change in flower color, but the change in flavonoid contents was not obvious. Additionally, the colorimeter analysis showed that as the flower color deepened in the seven materials, the ‘L’ value decreased from 92.46 to 62.32, indicating that the flower color gradually became darker. Similarly, the ‘a’ value declined steadily from −1.66 to −12.48, indicating a gradual deepening of the green color, while the ‘b’ value (which represents the yellow color) gradually increased from *C. vestitum* to *C. morifolium* ‘Greenlizard’, and then declined in *C. morifolium* ‘Green anna’ and ‘Lv Dingdang’ (Figure 2). These results suggest that chlorophyll is the main color-forming pigment in chrysanthemums with green flowers, and both chlorophyll and carotenoid affect the flower color in these materials.

### 3.2. Transcriptome Analysis of C. vestitum and C. morifolium Cultivars

*C. vestitum* is a wild species of chrysanthemum with pure white flowers. *C. morifolium* cultivars ‘Chunxiao’ and ‘Green anna’ produce green flowers; however, the green color of ‘Chunxiao’ flowers gradually fades away with the flower opening, whereas that of ‘Green anna’ flowers is very stable. RNA-seq of the outer-whorl ray florets of these three accessions (three biological replicates per accession; nine samples total) generated 423,277,980 raw reads. After the removal of adaptor and low-quality sequences, a total of 403,791,250 clean reads were obtained (Q30 = 89.78%–91.48%) (Table 1). After assembled with Trinity, a total of 262,601 unigenes were obtained (Appendix A), with the minimum and maximum lengths of 201 and 14,698 bp, respectively, and the N50 value of 1079 bp. Analysis of the DEGs showed that 19,764 unigenes were upregulated in ‘Green anna’ compared with ‘Chunxiao’, while 33,057 and 24,118 unigenes were upregulated in group ‘Green anna’ vs. *C. vestitum* and ‘Chunxiao’ vs. *C. vestitum*, respectively. Additionally, 1269 unigenes were upregulated in the above three groups (Figure 3, Appendix A). In addition, 14,910, 26,154, and 26,963 unigenes were downregulated in ‘Green anna’ vs. ‘Chunxiao’, ‘Green anna’ vs. *C. vestitum* and ‘Chunxiao’ vs. *C. vestitum* comparisons, respectively, and 990 unigenes were significantly downregulated in all three groups (Figure 3, Appendix A).

Furthermore, KEGG pathway enrichment analysis showed that in the ‘Green anna’ vs. *C. vestitum* comparison, a large number of upregulated genes were enriched in the photosynthesis, carotenoid biosynthesis, porphyrin and chlorophyll metabolism pathway; 94 unigenes were enriched in the photosynthesis pathway; 81 unigenes were enriched in the porphyrin and chlorophyll metabolism pathway (Appendix A).

Similarly, in the ‘Chunxiao’ vs *C. vestitum* comparison, a large number of upregulated genes were enriched in the ribosome, photosynthesis, chlorophyll and chlorophyll metabolism pathway, while 94 and 75 unigenes were enriched in the photosynthesis pathway and porphyrin and chlorophyll metabolism pathway, respectively (Appendix A). Photosynthesis and chlorophyll metabolism are closely related and may play important roles in the development of flower color in green chrysanthemum.

### 3.3. Expression Pattern of Chlorophyll Synthesis and Degradation Related Genes

Based on the KEGG analysis, 35, 7 and 4 chlorophyll biosynthesis, cycling and 4 degradation-related unigenes, respectively, were differentially expressed in the ‘Green anna’ vs *C. vestitum* and ‘Chunxiao’ vs *C. vestitum* groups (Appendix A).

The results of the qRT-PCR analysis showed that expression levels of chlorophyll synthesis related genes, including Cluster-35308.131599 (*HEMA1*), Cluster-35308.149260 (*CHLI1*), Cluster-35308.124019 (*CHLH1*), Cluster-35308.115727 (*CHLM1*), Cluster-35308.123158 (*CRD1*) and Cluster-35308.134839 (*PORA1*), varied significantly among the three accessions, consistent with the transcriptome data (Appendix A). By contrast, expression levels of genes related to chlorophyll cycling and degradation showed no significant differences among the three genotypes (Appendix A).

In ‘Chunxiao’, the color of ray florets in the outer whorl gradually changes from green to white during flower opening. Our results showed that the chlorophyll content of the outer-whorl ray florets also decreased gradually during flower opening, and the expression of chlorophyll synthesis related unigenes *HEMA1*, *CHLI1*, *CRD1* and *PORA1* also showed a steady decline (Figure 4).

It has been shown that ABA promotes the degradation of chlorophyll and causes the yellowing of leaves [27]. Here, we treated cut ‘Lv Dingdang’ flowers with 20 mg/L ABA and examined the expression of chlorophyll synthesis-related genes. The results showed that the expression of *HEMA1*, *CHLI1*, *C**HLH1*, *CRD1* and *PORA1* was significantly inhibited by ABA (Appendix A).

### 3.4. Selection of Key Transcription Factors involved in Chlorophyll Synthesis

Analysis of DEGs showed that 45 and 58 transcription factor-encoding genes were significantly upregulated and downregulated in the three comparison groups, ‘Green anna’ vs. ‘Chunxiao’, ‘Green anna’ vs. *C. vestitum*, and ‘Chunxiao’ vs. *C. vestitum* (Appendix A).

The expression of a subset of the up- or down-regulated transcription factor-encoding genes was analyzed by qRT-PCR. The results showed that the expression of 10 genes increased gradually in *C. vestitum*, ‘Chunxiao’ and ‘Green anna’, while that of 4 genes, such as Cluster-35308.188936 (*bHLH*), decreased gradually in the three materials, consistent with the transcriptome data. Among these genes, Cluster-35308.177654, a *NAC* family gene, showed the most significant change in expression (Figure 5).

The RNA-seq data of Cluster-35308.177654 (*NAC*) and chlorophyll synthesis-related genes were used to analyze their Pearson correlation coefficients with SPSS 22.0 (IBM, Armonk, NY, USA). The results showed that the expression of *HEMA1*, *CRD1*, *PORA1*, *CHLI1*, *CHLM1* were significantly correlated with Cluster-35308.177654 (*p* < 0.05), for *HEMA1*, *CRD1* and *PORA1*, the correlation was highly significant (*p* < 0.01) (Appendix A).

The expression of Cluster-35308.177654 (*NAC*) was further examined in the flowers of *C. morifolium* cultivars ‘Chunxiao’ (at different opening stages) and ‘Lv Dingdang’ (treated with or without ABA). In ‘Chunxiao’, the expression of Cluster-35308.177654 (*NAC*) increased significantly from stage 1 to stage 2 and then declined gradually from stage 2 to stage 4. In ‘Lv Dingdang’, ABA treatment significantly decreased the expression of Cluster-35308.177654 (Figure 6).

### 3.5. Characterization of CmNAC73

Our results showed that the expression of Cluster-35308.177654 (*NAC*) was highly correlated with chlorophyll synthesis (Figure 2, Figure 4, Figure 5 and Figure 6). To investigate the role of Cluster-35308.177654 in chlorophyll synthesis, we first cloned this gene from *C. morifolium* cultivar ‘Chunxiao’. The full-length CDS of Cluster-35308.177654 (*NAC*) is 876 bp in length and is predicted to encode a protein of 291 amino acids. Phylogenetic analysis revealed that the protein encoded by Cluster-35308.177654 (*NAC*) showed high homology to *NAC73* genes in many other species. Therefore, we named the Cluster-35308.177654 (*NAC*) gene as *CmNAC73* (Genbank accession: MW916171) (Appendix A).

Next, we transiently overexpressed *CmNAC73* in the leaves of *C. morifolium* cultivar ‘Lv Dingdang’. Leaves overexpressing *CmNAC73* exhibited delayed yellowing and significantly higher expression of *HEMA1*, *CHLI1*, *CHLM1*, *CRD1* and *PORA1* compared with leaves expressing the Super1300 empty vector control (Figure 7).

### 3.6. Regulation of Chlorophyll Synthesis-Related Genes by CmNAC73

Based on the published reference genome of chrysanthemum, the promoters of *HEMA1*, *CHLI1*, *CRD1* and *PORA1* (1648, 1245, 1580, and 1192 bp, respectively) were cloned. Transient transactivation assays in *N. benthamiana* leaves showed that CmNAC73 significantly induced the promoters of *HEMA1*, *CHLI1* and *CRD1* but did not have a significant effect on the promoter activity of *PORA1* (Figure 8a).

Previous studies have shown that NAC transcription factors bind to the core sequence CACG or CGTG [28,29]. In this study, 13, 5, 6 and 1 putative NAC-binding sites were found in the promoters of *HEMA1*, *CHLI1*, *CRD1* and *PORA1*, respectively (Figure 8b). To test whether CmNAC73 binds to the promoters of *HEMA1*, *CHLI1*, *CRD1* and *PORA1*, we performed a yeast one-hybrid assay. When *pHEMA1*-His and pGAD-CmNAC73 were co-transferred into yeast, the positive clone of yeast cells could grow normally on SD/-Leu-Trp-His, supplied with 4 mM 3-AT, while the yeast carrying pGAD and *pHEMA1*-His could not grow, which suggested CmNAC73 have a weak binding to the *HEMA1* promoter. Additionally, CmNAC73 showed strong binding to the promoter of *CRD1*, and 70 mM 3-AT can completely inhibit the expression of *HIS* gene in the combination of GAD + *pCRD1*, whereas CmNAC73 + *pCRD1* could grow normally (Figure 8c,d).

## 4. Discussion

Chrysanthemum is an important flower, owing to its diverse flower shapes, bright colors and a long ornamental period. Chrysanthemum has more than 30,000 cultivars [30]. Depending on the flower color, 811 varieties of chrysanthemum were divided into nine groups, including brown, orange, pink, purple, red, white, yellow, yellow-green and dark red; green varieties are very rare and are therefore classified into the yellow-green group [31]. It has been speculated that the ancestry of modern chrysanthemum includes *C. vestitum* (2n = 54), *C. indicum* (2n = 18, 36), *C. lavandulifolium* (2n = 18), *C. nankingense* (2n = 18) and *C. zawadskii* (2n = 54) [32]. Interestingly, none of these ancestral species produce green flowers. Moreover, the evolution of green chrysanthemums remains unclear.

The flower is an important organ of angiosperms derived from leaves [33]. One of the important functions of leaves is photosynthesis, which takes place in chloroplasts. These photosynthetic organelles have also been observed in some green flowers by transmission electron microscopy [4,34]. In transgenic Arabidopsis and *Nicotiana tabacum* lines overexpressing *SUPPRESSOR OF OVEREXPRESSION OF CONSTANS 1* (*SOC1*) or *SOC1-like*, heat-stress induced the biogenesis of chloroplasts and the formation of green petals [35]. In many plants, such as lily and chrysanthemum, flowers are green at the early stages of blooming, and their chlorophyll content decreases during the opening process [4,34,36]. This phenomenon was also observed in *C. morifolium* cultivar ‘Chunxiao’ (Figure 4). A previous study reported that white flowers are formed partly as a result of chlorophyll degradation [37]. It has been speculated that the maintenance of green color in green chrysanthemum may be related to the continuous synthesis and reduced degradation of chlorophyll. The results of the current study also showed that the content of chlorophyll and expression of chlorophyll synthesis-related genes increased steadily in *C. vestitum* and *C. morifolium* cultivars ‘Chunxiao’ and ‘Green anna’, which was closely related to the change in their flower color (Figure 2, Appendix A).

The chlorophyll biosynthesis and degradation pathways have been elucidated [3]. Chlorophyll metabolism is highly upregulated during biological processes, such as photomorphogenesis [38], leaf senescence [39], fruit ripening [40] and green flower production [41], and is influenced by many factors, including light, hormones and nutrient availability [42,43,44]. Under magnesium deficiency conditions, chlorophyll in old leaves is largely degraded, and nutrients are transported to young leaves to sustain limited growth [45,46]. Additionally, Hormones such as ABA, ethylene and salicylic acid (SA) significantly promote leaf senescence [7,47], whereas CTK, GA and auxin delay this process partly by promoting chlorophyll synthesis [48,49,50].

During ABA-induced leaf senescence, several *NAC* genes are induced, which inhibits the accumulation of chlorophyll, partly by promoting the expression of genes related to chlorophyll degradation [15,16]. Conversely, in rice (*Oryza sativa*), OsNAC2 accelerated leaf senescence by promoting ABA biosynthesis [51]. A few *NAC* genes have been shown to negatively regulate leaf senescence. Overexpression of a grape (*Vitis vinifera* L.) *NAC* gene, *DRL1*, in *N. benthamiana* inhibited ABA biosynthesis and significantly inhibited leaf senescence [18,52]. However, whether the genes regulate chlorophyll metabolism remains unclear. During the ripening process of citrus (*Citrus reticulata* Blanco) fruits, chlorophyll degradation was accompanied by fruit color changes, and two ethylene-responsive genes, *CitERF6* and *CitERF13*, promoted chlorophyll degradation, leading to fruit degreening [53]. Some studies have also reported the regulation of chlorophyll metabolism in green flowers. In chrysanthemum, CmCOL16, CmERF and CmbHLH transcription factors were found to be closely related to chlorophyll synthesis [4]. In petunia, overexpression of *PhCOL16a*, a homolog of the *CmCOL16* gene, significantly promoted the synthesis of chlorophyll [19]. In this study, we found a novel chrysanthemum *NAC* gene, *CmNAC73*, whose expression was repressed by ABA, and its transient overexpression in chrysanthemum leaves delayed leaf senescence by activating several chlorophyll synthesis-related genes, including *HEMA1*, *CHLI1*, *CHLM1*, *CRD1* and *PORA1* (Figure 6 and Figure 7). Furthermore, transient transactivation and yeast one-hybrid assays showed that CmNAC73 binds to the promoters of *HEMA1* and *CRD1*, thus activating their expression (Figure 8).

## 5. Conclusions

In short, we carried out transcriptome analysis of three chrysanthemum accessions, including *C. vestitum* and *C. morifolium* cultivars ‘Chunxiao’ and ‘Green anna’, with white, light-green and dark-green colored flowers, respectively, and identified a novel *NAC* gene, *CmNAC73*, which acts as a positive regulator of chlorophyll biosynthesis, at least in part, through the direct activation of chlorophyll synthesis-related genes *HEMA1* and *CRD1*. These results provide a better understanding of the formation of a green color in chrysanthemum flowers and the metabolism of chlorophyll in plants and guide the breeding of more green-flower chrysanthemum cultivars.

## Figures and Tables

**Figure 1 genes-12-00704-f001:**
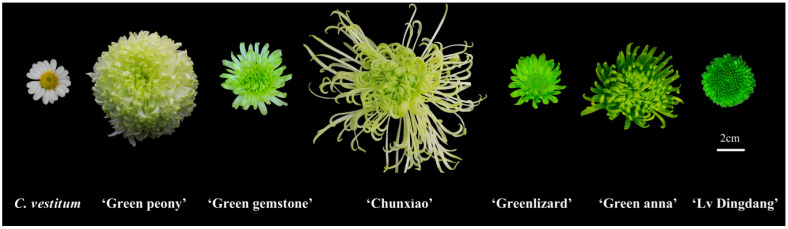
Phenotypic comparison of the flowers of *Chrysanthemum vestitum* (wild species) and six *C. morifolium* cultivars, ‘Green peony’, ‘Green gemstone’, ‘Chunxiao’, ‘Greenlizard’, ‘Green anna’ and ‘Lv Dingdang’ (from left to right). Scale bar = 2 cm.

**Figure 2 genes-12-00704-f002:**
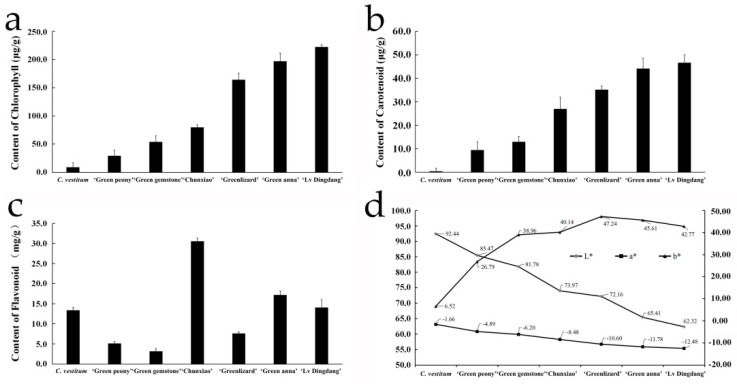
Pigment composition and colorimetry analysis of flowers of *C. vestitum* and six *C. morifolium* cultivars. (**a**–**c**) chlorophyll (**a**), carotenoid (**b**) and flavonoid (**c**) contents of flowers determined with a UV spectrophotometer. Data represent the mean ± standard error (SE) of three biological replicates. (**d**) Flower color determined with a colorimeter. The ‘L’ value indicates brightness (range: 0–100), while the ‘a’ and ‘b’ values indicate color (a > 0 represents the degree of red color, and a < 0 represents the degree of green; b > 0 represents the degree of yellow, and b < 0 represents the degree of blue).

**Figure 3 genes-12-00704-f003:**
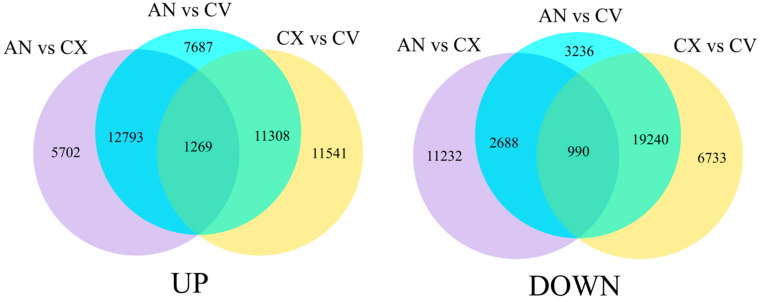
Venn diagrams showing the number of genes differentially expressed between *C. vestitum* (CV) and *C. morifolium* cultivars ‘Chunxiao’ (CX) and ‘Green anna’ (AN). The venn diagram on the left shows upregulated genes, and that on the right shows downregulated genes.

**Figure 4 genes-12-00704-f004:**
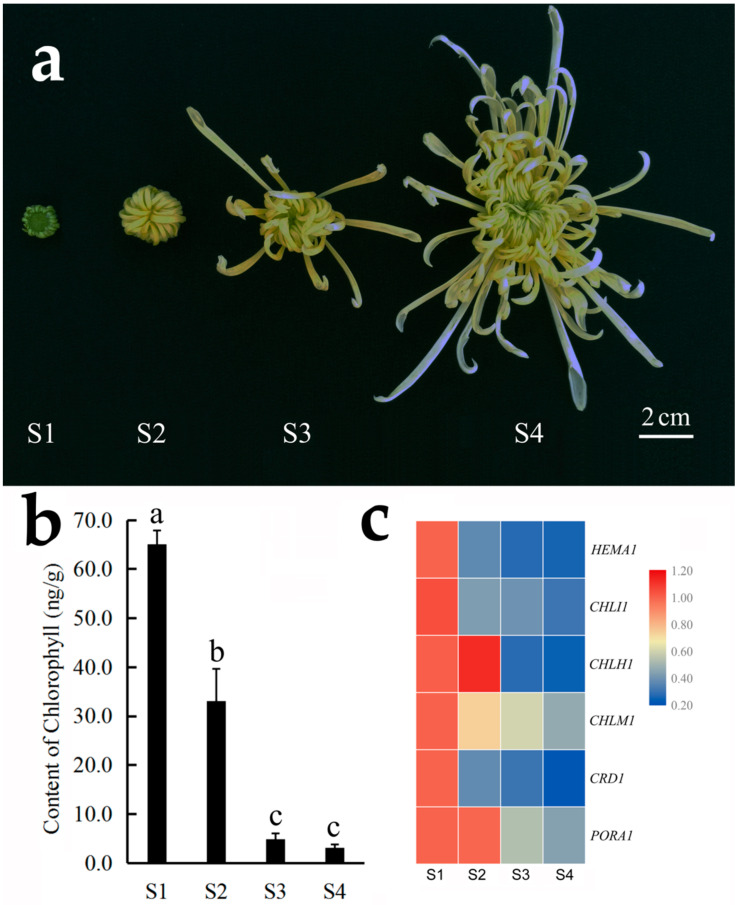
Content of chlorophyll and expression of chlorophyll synthesis related genes during flower opening in *C. morifolium* cultivar ‘Chunxiao’. (**a**) Different flower opening stages: stage 1 (S1), the outer-whorl ray florets are short, and therefore cannot wrap the inner disc florets; stage 2 (S2), inner disc florets are wrapped by the outer ray florets; stage 3 (S3), a few outer-whorl ray florets spread out; stage 4 (S4), more ray florets spread out, and the outer-whorl ray florets turn white. (**b**) Chlorophyll content. Data represent the mean ± SE of three biological replicates. (**c**) Heat map showing the expression of *HEMA1*, *CHLI1*, *CHLH1*, *CHLM1*, *CRD1* and *PORA1* at different stages of flower opening (S1–S4). The different colors represent gene expression levels determined by qRT-PCR analysis of three biological replicates.

**Figure 5 genes-12-00704-f005:**
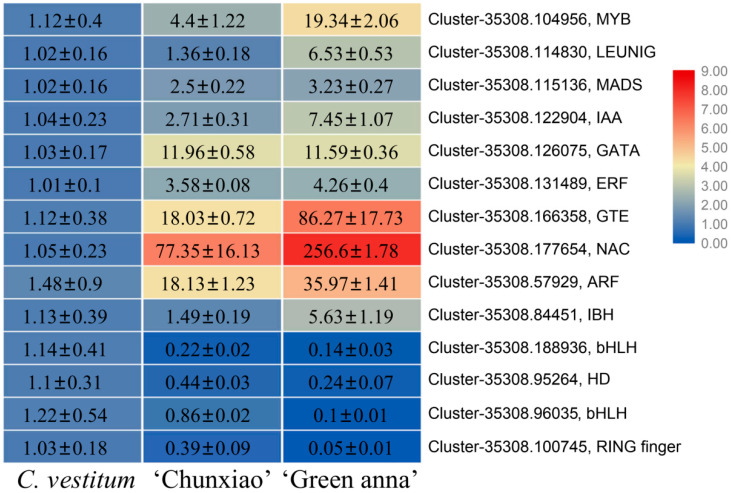
Expression of candidate transcription factor-encoding genes in *C. vestitum* and *C. morifolium* cultivars ‘Chunxiao’ and ‘Green anna’. The different colors represent gene expression levels determined by qRT-PCR. Data represent the mean ± SE of three biological replicates.

**Figure 6 genes-12-00704-f006:**
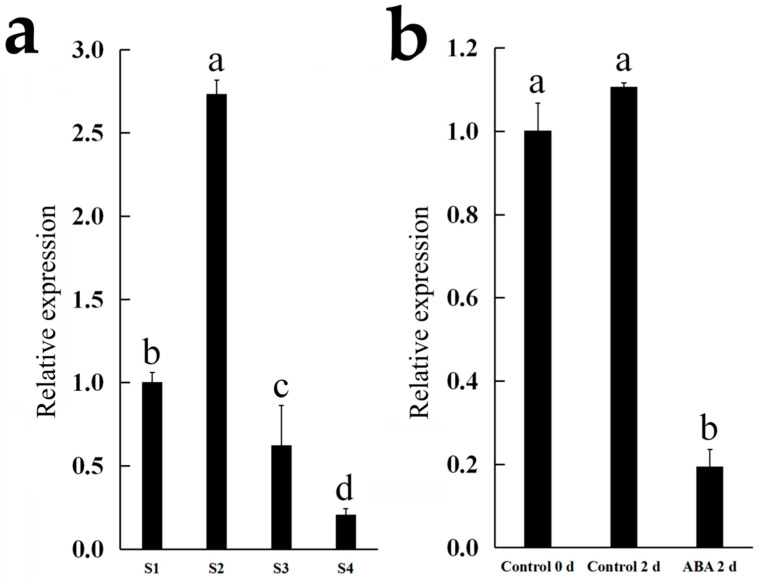
Expression analysis of Cluster-35308.177654 (*NAC*) by qRT-PCR. (**a**,**b**) Expression of Cluster-35308.177654 (*NAC*) in *C. morifolium* cultivar ‘Chunxiao’ flowers at different opening stages (**a**) and in cut flowers of *C. morifolium* cultivar ‘Lv Dingdang’ treated with or without 20 mg/L ABA. Data represent mean ± SE of three biological replicates.

**Figure 7 genes-12-00704-f007:**
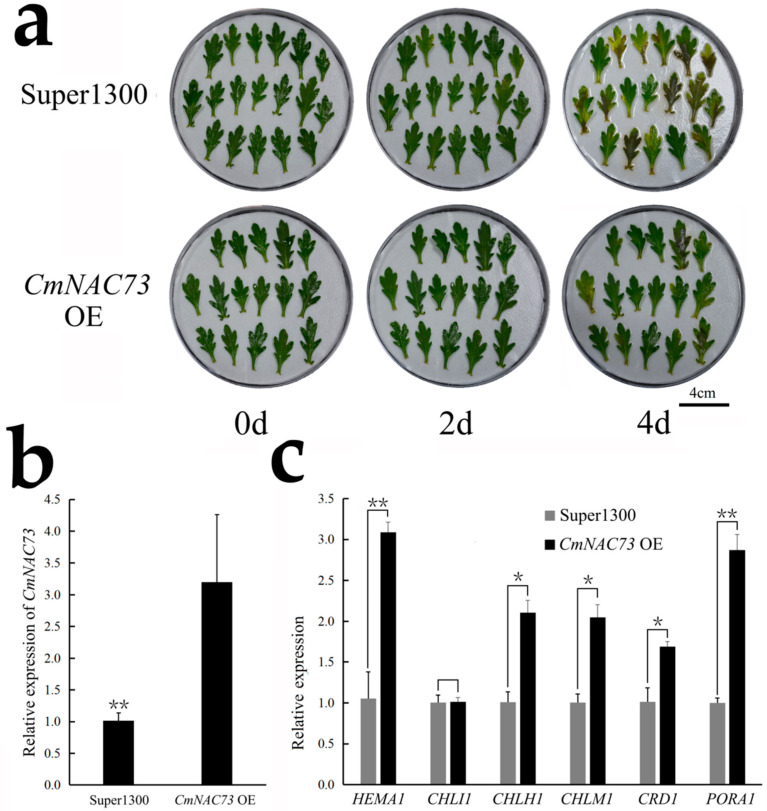
Transient overexpression (OE) of *CmNAC73* in the leaves of *C. morifolium* cultivar ‘Lv Dingdang’. (**a**) Phenotypic analysis of leaves transformed with the Super1300 empty vector and *CmNAC73*. (**b**,**c**) Expression analysis of *CmNAC73* (**b**) and chlorophyll synthesis related genes (**c**) in the control and *CmNAC73* OE samples by qRT-PCR. Data represent the mean ± SE of three biological replicates. For Student’s *t*-test, * *p* < 0.05; ** *p* < 0.01.

**Figure 8 genes-12-00704-f008:**
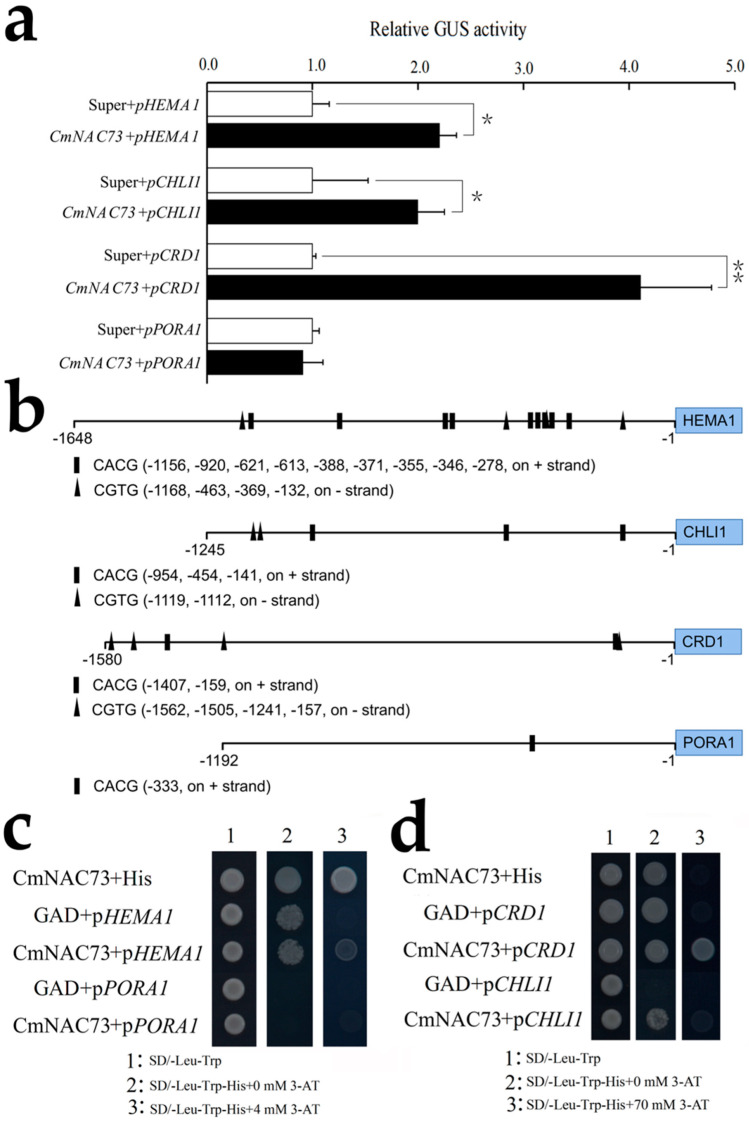
Regulation of *HEMA1* and *CRD1* by CmNAC73. (**a**) Transient transactivation assays of *HEMA1*, *CHLI1*, *CRD1* and *PORA1* promoters in *N. benthamiana* leaves. Data represent mean ± SE. Asterisks indicate significant differences (For Student’s *t*-test, * *p* < 0.05; ** *p* < 0.01). (**b**) Schematic representation of NAC-binding sites in the promoters of *HEMA1*, *CHLI1*, *CRD1* and *PORA1*. Squares and triangles represent binding sites on the + and - strands, respectively. (**c**) Yeast one-hybrid assay testing the interaction between CmNAC73 and the promoters of *HEMA1*, *CHLI1*, *CRD1* and *PORA1*. (**d**) The SD/-Leu-Trp medium was used to select positive yeast clones, and 3-AT was used to suppress background growth due to leaky *HIS3* expression in the pHIS2.1 reporter vector.

**Table 1 genes-12-00704-t001:** Summary of RNA-seq data of *C. vestitum* and two C. *morifolium* cultivars.

Sample	No. of Raw Reads	No. of Clean Reads	Clean Bases (Gb)	Error (%)	Q20 (%)	Q30 (%)	GC (%)
CV_1	47,547,792	45,740,146	6.86	0.02	96.58	91.48	42.38
CV_2	46,194,264	44,664,084	6.7	0.02	96.26	90.73	42.70
CV_3	45,414,078	43,364,316	6.5	0.02	96.33	90.77	41.86
CX_1	42,158,158	39,809,564	5.97	0.02	96.12	90.29	43.29
CX_2	47,895,570	45,269,660	6.79	0.02	96.20	90.51	43.27
CX_3	55,487,802	53,570,088	8.04	0.02	96.58	91.37	42.55
AN_1	47,448,436	44,975,644	6.75	0.02	95.94	89.91	42.82
AN_2	47,070,580	44,627,760	6.69	0.02	95.87	89.78	42.82
AN_3	44,061,300	41,769,988	6.27	0.02	95.92	89.89	42.53

CV, *C. vestitum*; CX, ‘Chunxiao’; AN, ‘Green anna’.

## Data Availability

RNA-seq data have been deposited in the National Center for Biotechnology Information (NCBI) Sequence Read Archive (SRA) database under the accession no. PRJNA721568.

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
