# Peer review of "CmNAC73 Mediates the Formation of Green Color in Chrysanthemum Flowers by Directly Activating the Expression of Chlorophyll Biosynthesis Genes HEMA1 and CRD1"

_genes, 2021, doi:10.3390/genes12050704_

Round 1
Reviewer 1 Report
The manuscript by Jing Luo, Huan Wang, Sijia Chen, Shengjing Ren, Hansen Fu, Ruirui Li, Caiyun Wang, entitled CmNAC73 mediates the formation of green color in chrysanthemum flowers by directly activating the expression of chlorophyll biosynthesis genes HEMA1and CRD1 provides a nice story encompassing the physiological aspects of flower color, transcriptomic analysis of different cultivars together with qRT-PCR verification of a number of genes, identification of a number of genes that could potentially regulate biosynthesis and degradation of chlorophyll in green flowers, along with identification of the CmNAC gene and further characterization of this gene in transient and yeast 1-hybrid assays. While much of the information lays hidden in the supplementary files (one example: comparison between transcriptome and qRT-PCR), the data is available for those that are interested in working with this species. The only thing that is missing in the supplement (should be in Table 2) are the actual sequences involved in the Cluster designations. Trinity tends to produce a number of partial and overlapping sequences, especially when different species/cultivars are used for the assembly, that eventually needs to be consolidated to identify the “real” mRNA’s/CDS’s.
Thus, aside from some minor editorial word usage in the text that could be clearer (e.g. “contents” – lines 227/228, versus “content” – line 231), this work is well put together, the title reflects that work that is presented and the work presented is solid.
Author Response
Thank you very much for your suggestions.
The sequences of unigenes involved in the Cluster designations have been deposited in the National Center for Biotechnology Information (NCBI) Transcriptome Shotgun Assembly (TSA) database, the accession number has not been assigned, however, the data can be acquired through the SRA project no. PRJNA721568 a few days later. Chrysanthemum vestitum is an wild Chrysanthemum species, which is also an important ancestor for modern Chrysanthemum cultivars. In this work, the clean reads from Chrysanthemum vestitum and Chrysanthemum morifolium cultivar ‘Chunxiao’ and ‘Green anna’ were assembled with RSEM (Li et al, 2011), which was used as the reference sequence. Subsequently, the clean reads of each sample were mapped to the reference sequence, and the mapping rates ranged from 62.75% to 67.10%. The mapped sequences were further used for differentially expressed genes (DEGs) analysis. In gene cloning, the sequence from the above three kinds of materials were almost the same, although there existed some single nucleotide polymorphism (SNP) sites, suggesting that it is viable to assemble the clean reads from Chrysanthemum vestitum and Chrysanthemum morifolium cultivar ‘Chunxiao’ and ‘Green anna’.
In line 231, the “content” was revised to “contents”.

Reviewer 2 Report
Comments for Authors,
In my opinion Authors presented valuable data related to chlorophyll biosynthesis regulation by NAC73 transcription factor in Chrysanthemum cultivars.
I have some minor comments that should be considered by Authors:
RT-PCR
Size of PCR product for tested and reference gene
Reason to choose the CmUbi as internal standard- citation of previous research or analysis using bestKeeper tool (Pfaffl et a. 2004)
Concentration of RNA and volume of samples for RT-PCR
Nucleic acid quantification; instrument and method.
How the integrity of RNA was assured?
Was the RNA digested with RNase free DNaseI to assure removing of genomic DNA remnants before qPCR?
Complete thermocycling parameters.
Software used to analyse RT-PCR results.
RNA seq
Concentration and volume of cDNA library
Cut-off values for low quality sequences.
Explain the choice of cutoff for log2FC
Gene cloning
Lines 164 and 170- which restriction site/sites?
GUS assay
Line 208; method of protein assay
Other:
Line 543; instead of Genes should be Gene.
Adding results of co-expression studies (Expression Angler or related software) using A. thaliana NAC73 (At4g28500) could support presented data by finding genes homologous to those found by Authors for example HEMA1, CRD1 or other related to chlorophyll biosynthesis in A. thaliana as co-expressed with A. thaliana NAC73 (Toufighi et al 2005).
The BAR and other Data Analysis Tools for Plant Biology (utoronto.ca)
Expression Angler (utoronto.ca)
Author Response
Thank you very much for your suggestions.
RT-PCR
- Size of PCR product for tested and reference gene
Answer: In quantitative Real-time PCR (qRT-PCR), the size of PCR products for tested and reference genes ranged from 151 to 251 bp.
- Reason to choose the CmUbias internal standard- citation of previous research or analysis using bestKeeper tool (Pfaffl et a. 2004)
Answer: In the previous research, the chrysanthemum UBIQUITIN gene (Genbank accession no. EU862325) was widely used as internal control gene (Wei et al. 2017; Xu et al. 2013; Yang et al. 2014).
- Concentration of RNA and volume of samples for RT-PCR
Answer: In qRT-PCR, 1 μg RNA of each sample was used to synthesize cDNA. The concentration of RNA ranged from 560.7 to 1261.6 ng/μl. The volume of RNA depends on its concentration, varied from 0.79 to 1.78 μl.
- Nucleic acid quantification; instrument and method.
Answer: The nucleic acid was quantified with nanodrop 2000 (Thermo Fisher Scientific, Waltham, MA, USA). This sentence was added to the methods, in line 133 and 134.
OD260nm/OD280nm and OD260nm/OD230nm indicates the purity of the RNA.
- How the integrity of RNA was assured?
Answer: The integrity of RNA was evaluated with agarose gel electrophoresis and Agilent 2100 (Agilent Technologies, Santa Clara, CA, USA). This sentence was added to the methods, in line 134 and 135. High quality RNA shows the clear 28s and 18s bands, the brightness of 28s is about two times of 18s. Agilent 2100 showed the RIN (RNA Integrity Number) of RNA samples, and the RIN of RNA samples in Chrysanthemum vestitum and Chrysanthemum morifolium cultivar ‘Chunxiao’ and ‘Green anna’ varied from 7.8 to 9.2, which suggested that the RNA quality is very good.
- Was the RNA digested with RNase free DNaseI to assure removing of genomic DNA remnants before qPCR?
Answer: The cDNA template was synthesized with the cDNA Synthesis SuperMix (AE311-03; TransGen Biotech, Beijing, China). The kit contains a gDNA remover enzyme, which was used to break down the genomic DNA.
- Complete thermocycling parameters.
Answer: The thermocycling program was as follows: 95 ℃ for 5 min, followed by 40 cycles of three steps (95 ℃ for 15 s, 60 ℃ for 30 s, and 72 ℃ for 30 s). In the end, there was a melting curve analysis: 95 ℃ for 15 s, 60 ℃ for 60 s, and 95 ℃ for 15 s, which was used to ensure the specificity of the amplified product. These sentences were added to the methods, in line 155 to 160.
- Software used to analyse RT-PCR results.
Answer: The qRT-PCR results were analyzed with LightCycler® 96 SW 1.1 software, which comes with the LightCycler 96 (Roche, Basel, Switzerland). The 2−ΔΔCt method (Livak et al. 2001) was used to calculate relative gene expression levels.
RNA seq
- Concentration and volume of cDNA library
Answer: The concentration of RNA varied from 398 to 978 ng/μl. After quality Inspection and purification, 1 μg RNA of each sample was used for cDNA library construction, the volume of RNA depends on its concentration.
- Cut-off values for low quality sequences.
Answer: In the raw reads, if a sequence contains more than 50% of nucleotides with a correct recognition rate≤ 99% (Q value < 20), it is considered as low-quality sequence, and will be filtered in the clean reads.
- Explain the choice of cutoff for log2FC
Answer: In previous studies, for analysis of differentially expressed genes (DEGs), the selection criteria was often defined as: fold-change (FC) > 2 or FC < 0.5, which also means log2FC>1 (Xu et al. 2013; Fu et al. 2021).
Gene cloning
- Lines 164 and 170- which restriction site/sites?
Answer: In the multiple cloning site (MCS) of pSuper1300 vector, there exists cutting sites Hind III, Apal I, Sma I, Sal I, Swa I, Spe I and Kpn I. Through cutting site analysis of CmNAC73 sequence, Hind III and Kpn I were used for vector construction. In vector PBI121, Hind III and BamH I were used for vector construction of HEMA1, CHLI1, PORA1 and CRD1 promoters. The restriction sites were added to the methods.
GUS assay
- Line 208; method of protein assay
Answer: The protein concentration was determined with the Bradford method (Bonjoch and Tamayo, 2001), which was added into the methods.
Other:
- Line 543; instead of Genes should be Gene.
Answer: Sorry for the mistake, it has been corrected.
- Adding results of co-expression studies (Expression Angler or related software) using A. thalianaNAC73 (At4g28500) could support presented data by finding genes homologous to those found by Authors for exampleHEMA1, CRD1or other related to chlorophyll biosynthesis in A. thalianaas co-expressed with A. thaliana NAC73 (Toufighi et al 2005).
The BAR and other Data Analysis Tools for Plant Biology (utoronto.ca)
Expression Angler (utoronto.ca)
Answer: Thank you for your advise, Expression Angler is a good tool for predicting gene co-expression. The name of CmNAC73 is based on its high homology with NAC73 of Tanacetum cinerarifolium, Lactuca sativa, and Helianthus annuus. However, the sequence similarity of CmNAC73 and ANAC073(AT4G28500) in Arabidopsis was only 45.81%, therefore, it might be not accurate to predict the co-expression of CmNAC73 with ANAC073(AT4G28500) in Arabidopsis. The Pearson correlation coefficients of CmNAC73 and chlorophyll synthesis-related genes were analyzed with SPSS 22.0 (IBM, Armonk, NY, USA). The results showed that the correlation of HEMA1 , CRD1, PORA1, CHLI1, CHLM1 and CmNAC73 were significant (P < 0.05), for HEMA1 , CRD1and PORA1, the correlation were highly significant (P < 0.01). This result was add to the main text, in line 341 to 346.

Reviewer 3 Report
Page 1, line 20, 22, 24, 26 please standardize the style throughout the article, sometimes the word “CmNAC73” is italicized
Page 1, line 44 please standardize the style ”glu-tRNA, Glu-tRNA”
Page 2, line 55, 59 please standardize the style, sometimes the word “SGR, SGR” is italicized
Page 4, line 175 not correct, two parentheses next to each other [MES] [pH 5.6]
Page 4, line 187 which means notation “at 5,000 ×g for 8 min”
Page 4, line 188 MgCl2,correct MgCl2
Page 6 Figure 2 d is difficult to read
Page 6 line 255 no spaces “...samples total)generated “
Question 1: The introduction should include: firstly, the current state of knowledge; second, the reason for the research; third, what the article is meant to do. On page 3 line 99-103, instead of answering the question why this study was undertaken, it is already stated that "...a novel NAC gene CmNAC73 was found to be closely related to chlorophyll synthesis ".
Question 2: Modern breeding is focused on the use of advanced biotechnological methods, including genetic transformation, allowing for direct achievement of the assumed goals, going beyond the pool of features conditioned by the genotype or bypassing cross-fertilization barriers. Since the novel NAC gene, CmNAC73, was detected in all chrysanthemums tested, can we assume that the green color of the chrysanthemums was obtained in the same way, e.g. by radiomutation?
Author Response
Thank you very much for your suggestions.
The point-by-point response were listed in the attachment.
